# Hydrothermal SiO_2_ Nanopowders: Obtaining Them and Their Characteristics

**DOI:** 10.3390/nano10040624

**Published:** 2020-03-27

**Authors:** Vadim Potapov, Roman Fediuk, Denis Gorev

**Affiliations:** 1Research Geotechnological Center, Far Eastern Branch of Russian Academy of Sciences, 30, Severo-Vostochny Highway, 683002 Petropavlovsk-Kamchatsky, Russia; vadim_p@inbox.ru (V.P.); denis.goreff2015@yandex.ru (D.G.); 2School of Engineering, Far Eastern Federal University, 8, Sukhanova Str., 690950 Vladivostok, Russia

**Keywords:** hydrothermal solution, polycondensation of orthosilicic acid, ultrafiltration membrane separation, cryochemical vacuum sublimation, diameter of SiO_2_ nanoparticles, nanopowder structure

## Abstract

The technological mode of obtaining amorphous SiO_2_ nanopowders based on hydrothermal solutions is proposed in this study. Polycondensation of orthosilicic acid as well as ultrafiltration membrane separation, and cryochemical vacuum sublimation were used. The characteristics of nanopowders were determined by tunneling electron microscopy, low-temperature nitrogen adsorption, X-ray diffraction, and small-angle X-ray scattering. The scheme allows to adjust density, particle diameters of nanopowders, specific surface area, as well as diameters, area and volume of the pore. Thus, the structure of nanopowders is regulated—the volume fraction of the packing of spherical particles in aggregates and agglomerates, the size of agglomerates, and the number of particles in agglomerates. The pour densities of the nanopowders depend on the SiO_2_ content in sols, which were 0.02 to 0.3 g/cm^3^. Nanoparticles specific surface area was brought to 500 m^2^/g by low temperature polycondensation. Nanoparticle aggregates specific pore volume (0.2–0.3 g/cm^3^) weakly depend on powders density. The volume fraction of the packing of SiO_2_ nanoparticles in aggregates was 0.6–0.7. Solid samples of compacted nanopowders had a compressive strength of up to 337 MPa. Possible applications of hydrothermal SiO_2_ nanopowders are considered.

## 1. Introduction

To date, a wide range of methods for producing various types of powders of amorphous dioxide silicon are known. At the same time, the need for SiO_2_ nanopowders—particles which have a high specific surface area up to 1000 m^2^/g and significant chemical activity—is increasing. Cheap sources of such materials and low-cost technologies for their production are needed.

Traditional applications of SiO_2_ nanopowders are known for the production of ceramics, glass, catalyst supports, sorbents, rubber fillers, polymeric materials, paper, abrasive materials, and medical preparations [1]. In the large-scale production of pyrogenic SiO_2_ nanopowders, the flame hydrolysis of SiCl_4_ in an atmosphere (H_2_-O_2_) is used [1]. The flame temperature, flow rate and volumetric proportions of SiCl_4_, H_2_, O_2_ gases, control the size and specific surface area of the nanoparticles. Another major production is the production of silica fume by condensation of gases in ferroalloy furnaces (condensed silica fume).

Another group of methods is based on the preparation of SiO_2_ particles from the liquid phase using a sol-gel transition. This group includes the preparation of SiO_2_ silicogels using a sol-gel transition followed by subcritical or supercritical gel drying [2,3]. In this case, the hydrolysis and polycondensation of molecules and the preparation of sols of colloidal particles of SiO_2_ are used at the first stage of the process. The precursors of SiO_2_ sols are metal alkoxides and chlorides, tetraethoxysilane and alkali metal silicates (Na, K, Li). At the gel stage, acid treatment with formamide is used to control the porous structure [4,5], and one of the most important parameters is the pH of the medium. The sol-gel method has produced a large number of mesoporous materials with a wide range of applications [6,7,8,9,10,11,12,13,14,15,16,17,18,19].

To obtain mesoporous materials, different variants of the Stober synthesis are used with the use of template additives of surfactants, water-soluble polymers, and previously obtained dense particles of sols [20]. Various forms of surfactant micellar solutions are used to synthesize mesoporous SiO_2_ particles [21,22,23,24,25,26,27,28]. SiO_2_ mesospheres are also synthesized with preliminary coagulation of the sol with electrolytes and subsequent polymer addition to separate the aggregates and prevent them from sticking together during drying [29]. There are methods for the synthesis of mixed oxides, hollow spheres, and objects of the core—mesoporous shell type [30,31]. SiO_2_ is one of the most common components for producing nanopowders, optical elements, medical preparations, thin films, fibers, nanotubes, nanowires, additives to hard films to increase tensile strength, hardness of hybrid coatings, and porous composite ceramics, SiO_2_-Me_x_O_y_nanocomposites [32,33,34,35,36,37,38,39,40,41,42,43,44,45]. The possibility of obtaining colloidal SiO_2_ based on cheap waste of glass powder was shown in [46]. The production of SiO_2_ powders from rice husk has been developed as well [47].

Hydrothermal solutions are a new raw material source for the production of SiO_2_ nanopowders. For its development, it is necessary to develop a technology for producing SiO_2_ nanopowders taking into account the parameters of the hydrothermal medium: temperature, pH, mineralization, ionic strength, polycondensation kinetics of orthosilicic acid, sizes and concentration of SiO_2_ particles, and stability of SiO_2_ nanoparticles in an aqueous medium.

The objectives of this article were:

- Create a technological route for the production of SiO_2_ nanopowders based on a hydrothermal solution with specific surface area up to 500 m^2^/g using the methods of ultrafiltration membrane separation and cryochemical vacuum sublimation.

- Create regulation parameters of the structure of the nanopowders: the diameters of SiO_2_ nanoparticles, specific surface area of nanopowders, diameters and specific pore volume, pour density, volume fraction of spherical particles in aggregates and agglomerates, sizes of agglomerates and number of particles in agglomerates.

- Assessment of possible applications of the obtained nanopowders.

## 2. Materials and Methods 

### 2.1. Methods for Producing Nanopowders

Silica is formed in a hydrothermal solution from molecules of orthosilicic acid (OSA), which comes from the chemical interaction of water of a hydrothermal solution with aluminosilicate minerals of rocks (orthoclase, microcline K(AlSi_3_O_8_), albite Na(AlSi_3_O_8_), anorthite Ca(Al_2_Si_2_O_8_, etc.) in the bowels of hydrothermal deposits at high pressures (10–25 MPa and above) and temperatures (250–300 °C and above). As the solution rises to the surface through the productive wells of geothermal power plants (GeoPP), temperature and pressure decrease, and the solution becomes supersaturated with respect to the solubility of *C_e_* amorphous silica. In the solution, polycondensation and nucleation of OSA molecules occur, leading to the formation of spherical silica nanoparticles with a diameter of 5 to 100 nm. In addition to silica, other components are in solution, the concentrations of which are given in Table 1. Silica is in solution in two states: solid (SiO_2_ particles) and dissolved (OSA molecules).

At the first stage of the process, OSA polycondensation and the growth of SiO_2_ nanoparticles were carried out at a certain temperature and pH of the hydrothermal solution. The final particle sizes of silica depend primarily on the temperature and pH at which the polycondensation of OSA molecules takes place. An increase in the polycondensation temperature and a decrease in pH slow down the reaction and increase the final particle size.

At the polycondensation stage, the temperature ranged from 20 to 90 °C (by preliminary cooling in heat exchangers), pH = 8.0–9.3. The range of silica concentrations in the initial solution is Ct = 400–800 mg/kg (t indicates the total silica content equal to the sum of the concentrations of the colloidal phase and dissolved Cs). The nucleation rate of silicic acid in an aqueous solution (nucl/(kg ∙ s)) is described by Equation (1) [48,49,50]: (1)$$I_{N}=Q_{LP}\times Z\times R_{MD}\times A_{cr}\times N_{A}\times M_{Si}^{-1}\times e^{-\Delta F_{cr}/k_{B}\times T},$$
where *Q*_LP_ = 3.34 × 10^25^ kg^−1^—the Lohse-Pound factor; *k_B_*—the Boltzmann constant; *M*_S__i_—the molar mass of SiO_2_; *N_A_*—the Avogadro number; *T*—the absolute temperature, K; *A*_cr_ = 4⋅*π* × *R*_c_^2^—critical nucleus surface area, m^2^; Δ*F*_cr_ = *σ*_s_ × *A*_cr_/3 = (16 × *π*/3) × *σ*_sw_^3^(*M*_SI_/*ρ* × *N*_A_ × *k*_B_×*T* × ln*S_m_*)^2^—change in free energy associated with the formation of a nucleus of critical radius *R*_c_; *ρ*—density of amorphous silica, kg/m^3^; *σ*_sw_—surface tension at the silica-water interface, J/m^2^; *Z*—Zeldovich factor.
(2)$$Z=\sqrt{-\frac{\partial^{2}\Delta F_{cr}/\partial n_{cr}^{2}}{2\times \pi \times k_{B}\times T}},$$
where *n_cr_ = (4* × *π/3)* × *(**ρ* × *N_A/_M_Si_)* × *R_c_^3^*—number of SiO_2_ molecules in the nucleus of critical size; *R_c_ = 2* × *σ_sw_* × *M_Si_/(**ρ* × *N_A_* × *k_B_* × *T* × *lnS_m_)*—critical radius; Z = (2/3) × (3/(4π × *ρ* × *n_cr_^2^))^1/3^* × *(**σ*_sw_/*k*_B_ × *T*)^0.5^; *R_MD_*—the rate of molecular deposition of silicic acid (g*⋅*cm^2^*⋅*min^−1^), which determines the particle growth rate: *R_MD_* = *F*(*pH*, *pH_nom_*) × *k_OH_*(*T*) × *f_f_*(*S_a_*) × (1 − *S_N_*^−1^),(3)
where k_OH_(T), F(pH, pH_nom_), *f_f_(S_a_)*—auxiliary functions depending on temperature, pH, ionic strength *I_s_* and supersaturation *S_m_*.

The characteristic polycondensation time—the temperature at which the supersaturation value decreased *e* = 2.71 times from the initial one—was at 20 °C and pH = 8.5, τ_p_ = 118.8 min, and at 50 °C, τ_p_ = 240.0 min.

With a decrease in the polycondensation temperature and an increase in the initial supersaturation *S_m_*, the nucleation *I_N_* rate increased and, accordingly, the final average diameter *d_m_* of SiO_2_ nanoparticles decreased, and the polycondensation of OSA passed faster. At pH = 8.0–9.3 and temperatures of 65–90 °C, the d_m_ values were 59–90 nm, at 40–65 °C, *d_m_* = 40–60 nm, and at 20–40 °C, d_m_ = 5–40 nm.

After completion of the polycondensation of OSA and the growth of SiO_2_ nanoparticles, concentrated aqueous sols were obtained by three-stage ultrafiltration membrane concentration. At the first stage, the SiO_2_ content in the sol was increased from 0.05 to 0.3–0.4 wt.%, at the second stage it increased up to 10 wt.%, on the third it increased up to 20% to 40 wt.%. The capillary type ultrafiltration membrane cartridge had an internal capillary diameter of 0.8 mm, a filter surface area of 55 m^2^, a minimum mass weight cut off parameter MWCO = 10–100 kD, a pressure drop across the membrane layer of 0.025–0.4 MPa, and permeability membranes (0.025–0.8) m^3^/m^2^·h·MPa. The final SiO_2_ content in sols was brought to 100.0–600.0 g/dm^3^ = 10–40 wt.%, salinity TDS = 800–2000 mg/dm^3^, specific conductivity 0.8–1.56 mS/cm, and dynamic viscosity 1–120 MPa·s (20 °C). The choice of pore sizes of polymer ultrafiltration membranes (MWCO = 10–100 kD) can provide high selectivity for SiO_2_ nanoparticles and low selectivity for ions of dissolved salts. Therefore, the parameter *m_s_* = [SiO_2_]/TDS continuously increases with increasing SiO_2_ content (up to 300 and higher), the inverse parameter (1/*m_s_*) decreases to 0.003 and lower, and there was no accumulation of ions in the concentrate. As a result, the value of the zeta potential of the surface of nanoparticles in concentrated sols fell in the range from −56 to −25 mV, which ensured the stability of particles to aggregation due to electrostatic repulsion without forced incorporation of stabilizers with SiO_2_ content up to 62.5 wt.%. Figure 1 shows the results of dynamic light scattering determination of the diameter distribution of particle volume for a sol sample with a content of SiO_2_ = 178 g/dm^3^, pH = 9.0, the average diameter of SiO_2_ particles in volume *d_m_* = 8.5 nm. The average value of the zeta potential of the particle surface found by the method electrophoretic light scattering was *ξ**_m_* = −42.0 mV (Zetasizer, Malvern, UK).

SiO_2_ nanopowders were obtained using cryochemical vacuum sublimation of sols. Cryochemical technology includes a sequence of main stages:

(1) Dispersion of the sol and cryocrystallization of droplets of a dispersed medium;

(2) sublimation of the solvent from the cryogranulate obtained in the previous step;

(3) desublimation of the solvent.

The cryochemical setup is shown in Figure 2 and Figure 3.

Before sublimation in a vacuum chamber, silica sols were dispersed using a nozzle, the droplets were solidified in liquid nitrogen at a temperature of 77 K, and cryogranules were obtained. After dispersion, the droplet size was 20 to 100 μm, the corresponding average droplet cooling rate was about 125 K/s, and the crystallization rate was 0.26 mm/s. The small size of the sol droplets and the high heat transfer surface made it possible to achieve rapid hardening of the droplets and the absence of particle adhesion. The particle sizes in the powders did not exceed the particle sizes in the sols. Vacuum sublimation took place at pressures from 0.02 to 0.05 mm Hg without fragments of droplet moisture and particles sticking together (Figure 4). To accelerate sublimation, heating was used. The temperature range of the heating surfaces in different parts of the vacuum chamber as it was heated during sublimation ranged from −80 to +25 °C (Figure 5). Productivity of the unit with a power consumption from 3 to 5 kW is 0.15–0.20 L/h. The residual water content in nanopowders was adjusted to 0.2 wt.%

### 2.2. Research Methods

The spherical shape of nanopowder particles was confirmed by TEM images obtained with a transmission electron microscope JEM-100CX, JEOL, Hiroshima, Tokyo, Japan.

Pour density of uncompacted nanopowders were measured using a PT-SV100 Scott volumeter (Pharma Test Apparatebau AG, Germany) with a system of alternating inclined shelves for “transfusion” of samples in the volume for weighing along a S-shaped path, which ensured the uniform distribution of nanopowders.

By the method of low-temperature nitrogen adsorption (ASAP-2010, Micromeritics Instrument Corporation, Norcross, GA, USA), adsorption-desorption curves were obtained. According to the adsorption-desorption curves for samples of nanopowders calculated the:

- BET area;

- specific surface areas, specific volumes, average diameters at one point and along the curves of adsorption and desorption of the area (BJH method);

- differential and integral distribution of area and volume over pore diameters;

- specific areas and volumes of micropores were found (with a diameter of less than 2 nm).

The amorphous structure of nanopowders was established by X-ray diffraction analysis (ARL X’TRA, Thermo Scientific, Basel, Switzerland).

Using small-angle X-ray scattering (SAXS), the dependences of the scattered radiation intensity function on the wave scattering vector were established (RigakuUltima IV, Rigaku Americas Corporation, Woodlands, TX, USA, rotating Cu anode, X-ray wavelength - 1.54 Å).

To determine the concentration of impurity components in nanopowders, a S4 PIONEER X-ray fluorescence spectrometer (Bruker, GmbH, Bremen, Germany) was used. Thermogravimetric analysis and estimation of mass losses during heating of the nanopowder were performed on a Pyris Diamond TG/DTA derivatograph (PerkinElmer LLC, Norwalk, CT, USA).

For determining compressive strength of the samples of compacted nanopowders servohydravlic machine Shimadzu AGS-X (Shimadzu Corporation, Japan) was used.

## 3. Results

### 3.1. SEM Images

According to scanning electron microscopy images with magnification factors equal to 250–7000, the sizes of the structures formed because of vacuum sublimation cryogranules of sols were within 20.0 to 100.0 μm. Figure 6 shows images of powder structures after sublimation of the solvent with a successively increasing coefficient of increase of 247, 500, 800, 1000, 2500, and 7000 times. After removal of the solvent, a porous-mesh structure of powder particles remains, preserving the features of a spherical shape and the size of solid cryogranule. Cavities formed inside the residual structures in their central part after solvent removal, which indicates the hardening mechanism of a sol drop of water removal from solid cryogranules. With light exposure, the residual structures were destroyed, forming flakes with a thickness of 0.1 to 0.2 μm.

Figure 7 shows the images of nanopowder particles obtained by transmission electron microscopy (TEM) (the content of [SiO_2_] in the sol before sublimation is 100 g/dm^3^).

### 3.2. Pour Density of SiO_2_ Nanopowders 

Table 2 and Figure 8 show the dependence of the pour density of powders ρ_p_ on the content of [SiO_2_] in sols. When the content of [SiO_2_] in sols was from 2.4 to 90 g/dm^3^, the density ρ_p_ after sublimation of water molecules from cryogranules and replacing them with air molecules was higher than the content of SiO_2_ in sols. Accordingly, after the sublimation of water molecules, the volume concentration of SiO_2_ nanoparticles in the nanopowder increased compared to the sol, and the average distance between the particles decreased. Therefore, at [SiO_2_] = 2.4 g/dm^3^, ρ_p_ was 20.5 g/dm^3^. When the content of [SiO_2_] in sols was higher than 90 g/dm^3^ after sublimation of water molecules, the concentration of SiO_2_ nanoparticles in the volume with air molecules decreased, and ρ_p_ became lower than [SiO_2_]: at [SiO_2_] = 520 g/dm^3^, ρ_p_ = 274 g/dm^3^. Accordingly, after sublimation of water molecules, the volume concentration of SiO_2_ nanoparticles decreased, and the average distance between particles increased. The ρ_p_/[SiO_2_] ratio decreased with increasing [SiO_2_] content in sols from 8.5 to 0.53, while in the range of [SiO_2_] = 100–520 g/dm^3^ it was changing relatively little: from 0.75 to 0.53. In the range of [SiO_2_] contents in sols from 100 to 520 g/dm^3^, the ρ_p_ ([SiO_2_]) dependence was close to linear.

### 3.3. Pore Characteristics of Nanopowders Obtained by Cryochemical Vacuum Sublimation of SiO_2_ Sols

Table 3 shows the characteristics of the pores of powder samples established by low-temperature nitrogen adsorption. The characteristics of the samples are given in order of increasing values of their BET area S_BET_. Figure 9, Figure 10 and Figure 11, for five of the samples in Table 3, in an ascending order of S_BET_, show graphs of nitrogen adsorption-desorption isotherms, differential and integral distributions of pore area and volume over diameters.

Nitrogen sorption-desorption isotherms are of type IV and have a hysteresis loop characteristic of mesopores with diameters from 2 to 50 nm and allow one to estimate the pore size distribution. Hysteresis on the isotherm graph allows us to conclude that nanopowders are a globular system consisting of spherical particles, each of which is in contact with two or more neighboring particles. By lowering the temperature of the hydrothermal solution at the OSA polycondensation stage from 90 to 20 °C, a decrease in the sizes of SiO_2_ particles was achieved. Additionally, there was an increase in their specific surface area and a decrease in the average pore diameter;
*d_p_* = *4 × V_p_/S_BET_*(4)

With a temperature decrease at the OSA polycondensation to 20 °C, the BET–nanopowder area was regulated and increased to 500 m^2/^g. In this case, the specific pore volume *V_p_* was in a narrow range of 0.20 to 0.30 cm^3^/g and the average pore size decreased to 2.7 nm (Table 3). The specific pore volume *V_p_* depended weakly on the density of nanopowders.

The specific pore volume *V_p_* = 0.20–0.30 cm^3^/g showed that spherical SiO_2_ particles form aggregates with a high-volume fraction. The volume fraction of SiO_2_ particles with a density of 2.2 g/cm^3^ in aggregates at *V_p_* = 0.20–0.30 cm^3^/g was *V_s_/V_aggr_* = 0.7–0.6 (*V_s_* is the volume of SiO_2_ particles occupied in the aggregate, *V_aggr_* is the aggregate volume), the density of the substance in aggregates was 1.32 to 1.54 g/cm^3^. The density of the substance in the aggregates was much higher than the density of nanopowders *ρ_p_* = 0.02–0.274 g/cm^3^.

The ratio of the average pore diameter *d_p_* to the average surface particle diameter *d_BET_* for most of the nanopowder samples ranged from 0.3 to 0.43, to 0.5 (Table 3), which also testified to the high-volume density of the packing of SiO_2_ particles in the aggregates. The differential distributions of the pore area and volume over the diameters are rather narrow and are characterized by a relatively small width. The fraction of micropore area in the studied nanopowders is no more than 10% to 15%, and the proportion of micropore volume is not more than 1% to 3% (Table 3).

The samples NM-200, 201, 204 of SiO_2_ nanopowders were produced by precipitation from precursor Na_2_SiO_3_ and the samples of pyrogenic SiO_2_ nanopowders were produced by the flame hydrolysis of SiCl_4_ [1]. Nitrogen sorption-desorption isotherms of precipitated samples NM-200, NM-201 and of pyrogenic SiO_2_ nanopowder NM-202 were another type then of hydrothermal nanosilica powdes (Figure 9, Figure 10, Figure 11 and Figure 12). Pore characteristics of pyrogenic and precipitated SiO_2_ nanopowders established by BET-method are in Table 4. The form of the hysteresis loop of NM-200, 201, and 202 samples differs from the form of loop of hydrothermal samples, and the structure of SiO_2_ particles aggregates and agglomerates differs in precipitated and pyrogenic samples from the hydrothermal samples. 

The specific pore volume *V_p_* = 0.499–0.513 cm^3^/g of pyrogenic SiO_2_ nanopowder NM-202, 203 showed that the volume fraction *V_s_/V_aggr_* of SiO_2_ particles in aggregates was about 0.5. In the samples of precipitated nanopowders NM-200, 201, 204 with *V_p_* = 0.79, 0.581, 0.50 volume fraction was *V_s_/V_aggr_* = 0.364, 0.438, 0.475. The volume fraction *V_s_/V_agg_* in the samples NM-(200-204) was lower than in the UF samples of hydrothermal nanopowders and indicated another structure of aggregates. The fraction of the area of the micropore and volume were the same as in UF samples. 

### 3.4. The XRD Data and Small Angle X-ray Scattering 

Samples of nanopowders had an amorphous structure without the presence of crystalline phases (Figure 13a). After calcination at 1200 °C for 2 h, cristobalite peaks appeared in the diffractogram of the samples (Figure 13b). In the X-Ray data of all samples NM-200, 201, 204 precipitated from Na_2_SiO_3_ precursor the presence of Na_3_SO_4_ crystalline impurities at 2Ө = 32, 34 degrees and crystalline impurities of Boehmite (γ-AlO(OH)) were observed [1]. In the pyrogenic samples NM-202 and 203, the presence of Boehmite was detected by XRD [1]. 

Samples of SiO_2_ nanopowders isolated from sols were studied by small angle X-ray scattering (SAXS) (Figure 14). The dependences of the intensity of the scattered electromagnetic radiation I_SR_ (q) on the wave vector *q = 4π* × *sin (Ө)/λ* (*Ө* is half the scattering angle and *λ* is the X-ray wavelength) were obtained for five different samples of SiO_2_ nanopowders in logarithmic coordinates. Sample 1: nanopowder obtained by cryochemical vacuum sublimation of the sol with a SiO_2_ content of 100 g/dm^3^ (precursor: hydrothermal solution). Samples 2 and 3: for nanopowders obtained by sol-gel and cryochemical vacuum sublimation of gels, the precursor is an aqueous solution of sodium silicate, the SiO_2_ content in the sol is 100 g/dm^3^. Sample 4: for nanopowder obtained by sol-gel transition and cryochemical vacuum sublimation of the gel, the precursor is an hydrothermal solution, SiO_2_ content in the sol 100 g/dm^3^. Sample 5: for nanopowder obtained by sol-gel transition and gel drying, the precursor is tetraethoxysilane.

According to Figure 14, only for sample 1 graph *logI_SR_(q) - log (q)* in the range *q* = 0.21 to 0 nm^−1^ was close to linear, which indicates the mode of scattering by fractal agglomerates [51,52,53,54,55,56,57]: I_SR ~_ q^-Df^, (5)
where *D_f_* is the fractal dimension. According to the slope of the dependence *logI_SR_(q)–log(q)*, the dimension *D_f_* for the nanopowder of sample 1 was 2.21. In the range *q* = 1.0 to 3.0 nm^−1^ for sample 1, the modulus of the slope of the *logI_SR_(q)–log(q)* dependence was 4.05, in the region *q* = 0.08 to 0.2 nm^−1^, it was 3.97, which corresponds to Porod’s scattering regime. For sample 1, approximation of the dependence *logI_SR_(q)* by the Guinier’s function is *I_SR_(q) = exp (-R_g_^2^*×*q^2^/3).* Here, *R_g_* is the gyration radius, for the ranges *q* = 1.0 to 3.0 nm^−1^ and *q* = 0.08 -to 0.2 nm^−1^, respectively, where the primary particle size is *2R_g1_* = 4.92 nm and the gyration radius of agglomerates is *2R_g2_* = 24.4 nm. The relation between gyration radius *R_g2_* and outer diameter of agglomerates outer diameter *D_agglom_* is *D_agglom_* = *((D_f_ +2)/D_f_)^0.5^*×*2R_g2_* = 33.7 nm. The number *N_agglom_* of primary SiO_2_ nanoparticles with a diameter of *2R_g1_*, which are in the fractal agglomerate of size *D_agglom_,* can estimated as [54,55]:N_agglom_ = (D_agglom_/2×R_g1_)^D^_f_ = 69.3 ~ 69-70.(6)

The average volume fraction of SiO_2_ primary particles in agglomerate is *(D_agglom_/2* × *R_g1_)^D^_f_^−3^ = 0.218.*

For sample 4, the scattering mode on *I_SR_ ~ q^-D^_f_* fractals was realized in the range q = 0.2 to 0.5 nm^−1^. The fractal dimension, determined by the slope of the dependence *logI_SR_(q)– log(q)*, is *D_f_* = 2.33.

Nanopowders obtained by cryochemical vacuum sublimation of sols based on a hydrothermal solution were characterized by a fractal dimension in the range *D_f_* = 2.04 to 2.21. Sodium silicate and tetraethoxysilane nanopowders were characterized by *D_f_* = 2.2 to 2.3.

Sample NM-202 of pyrogenic SiO_2_ powder obtained by the flame hydrolysis of SiCl_4_ was characterized by SAXS, TEM, DLS and BET methods in [1]: *D_f_* = 2.5, *2*×*R_g1_* = 16 nm, *2*×*R_g2_ =* 100 nm, *N_agglom_* = 200. Two samples, NM-200 and NM-201, were produced by precipitation from the precursor solution of Na_2_SiO_3_ were characterized by parameters: NM-200—*D_f_* = 2.45, *2*×*R_g1_* = 18 nm, *2*×*R_g2_ = 440 nm, N_agglom_* = 3500; M-201—*D_f_* = 2.45, *2*×*R_g1_* = 20 nm, *2*×*R_g2_ = 80 nm, N_agglom_* = 457. Fractal dimension of the hydrothermal nanopowders samples was lower than of the precipitated and pyrogenic samples [1,52,54,57,58]. Physicochemical and biophysicochemical interactions of nanoparticles with cells are in strong dependence from fractal dimension *D_f_* and parameters of the structure of agglomerates *2*×*R_g1_, D_agglom_, N_agglom_*, as from particles shape, surface electric charge and morphology [59,60,61,62,63,64].

### 3.5. The Limits of the Content of Impurity Components in Nanopowders

Table 5 shows the concentrations of impurity components in the silica nanopowder obtained by cryochemical vacuum sublimation of the sol at a SiO_2_ content of 500 g/dm^3^ in the sol. The total content of impurities with respect to SiO_2_ does not exceed 0.3 wt.%.

### 3.6. Evaluation of the Density of Surface Silanol Groups of Si-OH

Table 6 shows the dependence of the mass of the nanopowder sample (wt.%) on temperature, according to thermogravimetric analysis.

Taking into account the specific surface area of silica S_BET_ (m^2^/g) and the mass loss ΔmH_2_O (wt.%) due to the removal of water and OH-groups during thermogravimetric analysis, one can find the total concentration δ_OH_(OH/nm^2^) of all silanol groups. These groups are located both on the surface and in the volume of silica conventionally assigned to the specific surface of the nanopowder sample [65]: δ_OH_ = (Δm_H2O_ 2 × 6.02 × 10^3^)/(18 × S_BET_). (7)

Having taken S_BET_ = 300 m^2^/g for the sample, the final temperature at which all silanol groups are completely removed is equal to 1000 °C, and taking into account the data in Table 5, the values of the total *δ**_OH_* (on the surface and inside the volume) were obtained. These values conventionally calculated per unit surface area of the sample for different temperatures (Table 7).

Note. Symbol *T,* °C—temperature of sample pretreatment in vacuum. *δ_OH_* is the total water loss obtained by thermogravimetric analysis when the sample was calcined to high temperatures and expressed as the number of OH- groups, referred to the surface unit of SiO_2_. *α_OH_* is the averaged total true concentration of silanols on the SiO_2_ surface depending on the pretreatment temperature obtained by Zhuravlev according to the method of deutero-exchange [66]. *γ_OH_* is the content of internal silanols per unit surface area of SiO_2_, obtained as the difference between the corresponding *δ_OH_* and *α_OH_* values at the same fixed temperature (this value is also formally expressed as the number of OH groups per unit surface area of SiO_2_ (*γ_OH_*, OH/nm^2^)).

### 3.7. Experiments with Compacted SiO_2_ Nanopowders

Samples of SiO_2_ nanopowder were compacted on a hydraulic press at pressures of 1000 to 2000 MPa for 2 to 24 h; then, after hardening, they calcined at temperatures of 700, 800, 1000, and 1100 °C for 2 to 4 h. After compaction and calcination, the mechanical characteristics of solid samples were determined using the Shimadzu complex with registration of the force–strain curves (Figure 15 and Figure 16, Table 8). Table 8 shows the values of compressive strength in the range 135–337 MPa. This indicates a high specific surface and high surface energy of SiO_2_ nanoparticles.

Sizes of the sample 5 (thickness × width × height), mm: 5.0 × 11.9 × 2.9; sample density 1.7 g/cm^3^; indentation speed1 mm/min; maximum power 20057.6 N; maximum strain 337.1 N/mm^2^; amplitude of the stroke, 1.681 mm; maximum elongation, 1.681 mm; maximum deformation 57.88%; maximum time 100.95 s.

## 4. Prospects for Research and Applications of Hydrothermal Nanopowders SiO_2_

Further studies of the characteristics and possible applications of hydrothermal SiO_2_ nanopowders can be continued in the following areas:

- production of silicates of metals [67,68,69];

- receiving glasses;

- obtaining silicon carbide SiC;

- formation of ceramic forms based on SiO_2_ nanopowders;

- obtaining heat insulators;

- determination of the sorption capacity of nanopowders and obtaining sorbents for water purification and sorbents for gas chromatography;

- studies of the possibility of using nanopowders as catalyst supports.

Using SiO_2_ nanoparticles, which have a high and chemically active surface, one can purposefully influence [68,69,70,71,72,73,74,75,76,77,78,79]:

- the kinetics of hydration of the basic cement minerals C_3_S, C_2_S, C_3_A, C_4_AF and increasing the rate of CSH gel formation up to 20% and polymerisation [68,69,77,78];

- reducing the size and shape of the particles of the gel of the hydrates of calcium silicate C-S-H, increasing the density of their volume packaging;

- reducing content of Ca(OH)_2_ up to 20% to 30% and, thus, increasing content of CSH gel in hardened concrete because of rapid kinetics of pozzolanic reaction of SiO_2_ nanoparticles with Ca(OH)_2_ [74,75]; hydrothermal SiO_2_ nanoparticles with great specific surface area up to 500 m^2^/g and high chemical activity due the surface density of Si-OH groups up to 4.9 nm^−2^, which significantly accelerates the kinetics of pozzolanic reaction [68,77,78];

- increase the volume fraction of C-S-H gel phases with greater elasticity and hardness, Ca/Si relation due to modification of nanostructure of hardened concrete, and, as a result, increase the compressive and bending strength of concrete, reduce pore volume, increase water resistance, frost resistance, chemical resistance, and, as a result, the durability of concrete.

Nanosilica obtained based on a hydrothermal solution is applicable as an effective nanomodifier of concrete and is used [77,78,79]:

(1) to accelerate hardening;

(2) increasing the compressive strength of concrete at the early age up to 120% and about 40% in the age of 28 days; increasing of the concrete’s compressive strength with additive of hydrothermal nanosilica was 10% higher than with additive of colloid nanosilica based on Na_2_SiO_3_ precursor [72]; 

(3) reduction of Portland cement consumption up to 30%.

A sufficiently developed application of hydrothermal SiO_2_ nanoparticles is the intensification of photosynthesis in chloroplasts of plant cells due to the photoluminescent radiation of SiO_2_ nanoparticles. SiO_2_ nanoparticles due their optical properties can absorb solar radiation in ultraviolet region with a wave-length of 200 to 360 nm and emit of luminescent radiation in visible region with a wave length of 400 to 500 nm, in which the efficiency to absorb radiation by photosynthetic pigments and carotenoids is high [80,81,82]. An increase in the proportion of photosynthetic pigments of chlorophylls a (62%) and b (79.3%) [82,83,84], as a result, an increase in the growth rate, biochemical and biometric indicators at all stages of plant growth and development, a significant increase crop yields of agricultural plants from 9% to 60% [82,83,84,85], increase of contents of carotenoids—14.5%, B_2_—130%, B_5_—60%, B_6_—230%, B_9_—230% and C—14.4% vitamins [82,83,84] and rising biological activity of raw plant’s mass with respect to cultures of Daphnia magna—352% and Paramecium caudatum—90.5% [82,83,84,85,86]. Hydrothermal SiO_2_ nanoparticles have great inhibition ability on microflora (Leveilluia taurica, Ocidiopsis sicula) [87]. 

Hydrothermal nanosilica used as a feed additive that increases the productivity of farm animals (8% to 10%), rate of mass growth (10% to 40%), strength of the bone (17%), blood characteristics (Ca/P relation) and immunity (25% rising of the proportion of big forms of lymphocytes) [88,89,90,91].

Non-toxic [86,87,88,89,90,91] hydrothermal SiO_2_ nanopowders can be the basis for the creation of medical preparations:

- enterosorbents,

- drugs that improve the structure of bone, strength and plasticity of the articular-bone tissue and reduce Ca leaking.

## 5. Conclusions

1. A technological route proposed that allows one to obtain amorphous SiO_2_ nanopowders based on a hydrothermal solution. The scheme includes the OSA polycondensation processes, ultrafiltration membrane separation, and cryochemical vacuum sublimation. The route allows to regulate parameters of the structure of the powder: the pour density, the diameters of the particle, specific surface area, diameters, pore area and volume, volume fraction of spherical particles in aggregates and agglomerates, sizes of agglomerates and number of particles in agglomerates, and fractal dimension. The parameters of the structure of hydrothermal nanosilica powders (*ρ_p_, d_BET_, S_BET_, V_p_, V_s_/V_aggr_, 2*×*R_g1_, D_agglom_, N_agglom_*, *D_f_)* differs from precipitated and pyrogenic samples. The structure parameters determine physical and chemical activity and applications of nanopowders. The interactions between SiO_2_ nanoparticles, surface properties, parameters of double electric layer and stability of SiO_2_ nanoparticles differs in hydrothermal sols and nanopowders from interactions in sols produced from Na_2_SiO_3_ solutions or in precipited and pirogenic SiO_2_ nanopowders. The difference in interactions of SiO_2_ nanoparticles arised from the ion concentrations, ionic strength of hydrothermal solution and kinetics of OSA’s polycondencation. The difference in nanoparticles interactions leads to the difference in structure parameters of nanopowders. The structure parameters determines physical and chemical activity of SiO2 nanopowders and it’s applications.

The interactions between SiO_2_ nanoparticles, surface properties, parameters of double electric layer and stability of SiO_2_ nanoparticles differs in hydrothermal sols and nanopowders from interactions in sols produced from Na_2_SiO_3_ solutions or in precipited and pirogenic SiO_2_ nanopowders. The difference in interactions of SiO_2_ nanoparticles arised from the ion concentractions, ionic strength of hydrothermal solution and kinetics of OSA’s polycondencation. The difference in nanoparticles interactions leads to the difference in structure parameters of nanopowders. The structure parameters determines physical and chemical activity of SiO_2_ nanopowders and it’s applications.

2. The values of the average particle diameter of SiO_2_ in sols, according to the data of dynamic light scattering, ranged from 5 to 100 nm. The average particle diameter of SiO_2_ in powders, according to tunnel electron microscopy and the BET method, was in the same range of 5 to 100 nm.

3. The pour density of nanopowders *ρ_p_* depended on the content of [SiO_2_] in the sol and, therefore, on the concentration of particles and the average distance between them. When the content of [SiO_2_] in sols ranged from 2.4 to 90 g/dm^3^, the pour density was higher than [SiO_2_], respectively SiO_2_ particles came together after sublimation of water molecules. At a content of [SiO_2_] above 90 g/dm^3^, the pour density was lower than [SiO_2_] and the average distance between SiO_2_ particles, respectively, increased. In the range of [SiO_2_] contents in sols from 100 to 520 g/dm^3^, the *ρ_p_* ([SiO_2_]) dependence was close to linear.

4. By lowering the temperature of the hydrothermal solution at the OSA polycondensation stage from 90 to 20 °C, we achieved a decrease in the size of SiO_2_ particles and, accordingly, an increase in their specific surface to 500 m^2/^g and a decrease in pore diameter from 15 to 2.7 nm. The specific pore volume was in the range of 0.20 to 0.30 cm^3^/g and varied little depending on the specific surface area and density of nanopowders. Spherical particles of SiO_2_ in nanopowders form aggregates with a high-volume fraction of 0.7 to 0.6. The density of matter in aggregates, 1.32–1.54 g/cm^3^, was significantly higher than the density of nanopowders, which was 0.02–0.274 g/cm^3^. The ratio of the average pore diameter *d_p_* to the average surface particle diameter *d_BET_* from 0.3–0.43 to 0.5 also indicated a high-volume fraction of the packing of SiO_2_ particles in the aggregates. According to the SAXS data, aggregates of SiO_2_ nanoparticles form agglomerates with a fractal dimension of 2.04–2.21.

5. The content of impurity components in nanopowders can be brought up to 0.3 wt.% due to ultrafiltration membrane separation of SiO_2_ nanoparticles and ions of dissolved salts, an increase in the SiO_2_ content in the sol, and an increase in the ratio m_s_ = [SiO_2_]/TDS to 300 and higher.

6. Tests of compacted SiO_2_ nanopowders showed values of compressive strength in the range 135–337 MPa. This indicates a high specific surface area and high surface energy of SiO_2_ nanoparticles.

7. Nanopowders obtained by the proposed technology have prospects for the use in the production of glass, silicon carbide, ceramics, concrete nanomodifiers, sorbents, plant growth stimulants, feed additives for agricultural animals, and medicines.

## Figures and Tables

**Figure 1 nanomaterials-10-00624-f001:**
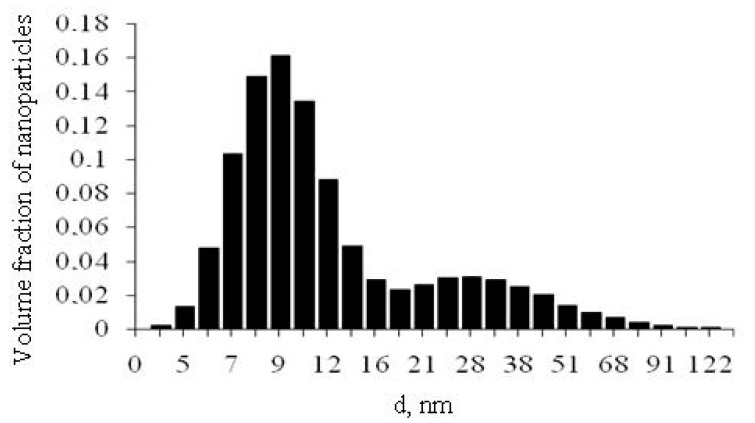
The particle sizes distribution in the hydrothermal sol sample, determined by dynamic light scattering.

**Figure 2 nanomaterials-10-00624-f002:**
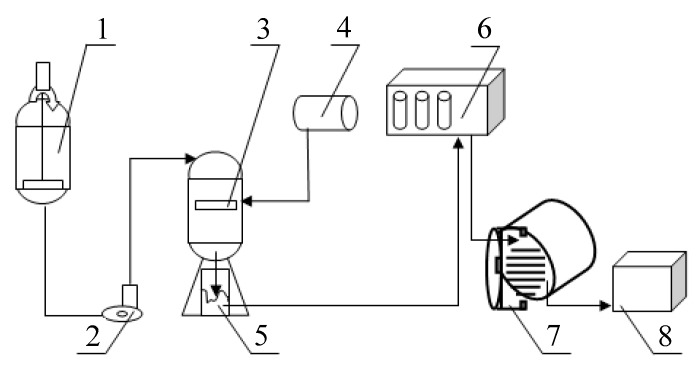
Scheme of the cryochemical vacuum sublimation setup for producing SiO_2_ nanopowder: 1—apparatus for the preparation of an aqueous sol; 2—metering pump; 3—cryogranulator; 4—tanker with liquid nitrogen; 5—capacity for storing cryogranules; 6—refrigerator; 7—sublimation apparatus; 8—box for storing nanopowder samples.

**Figure 3 nanomaterials-10-00624-f003:**
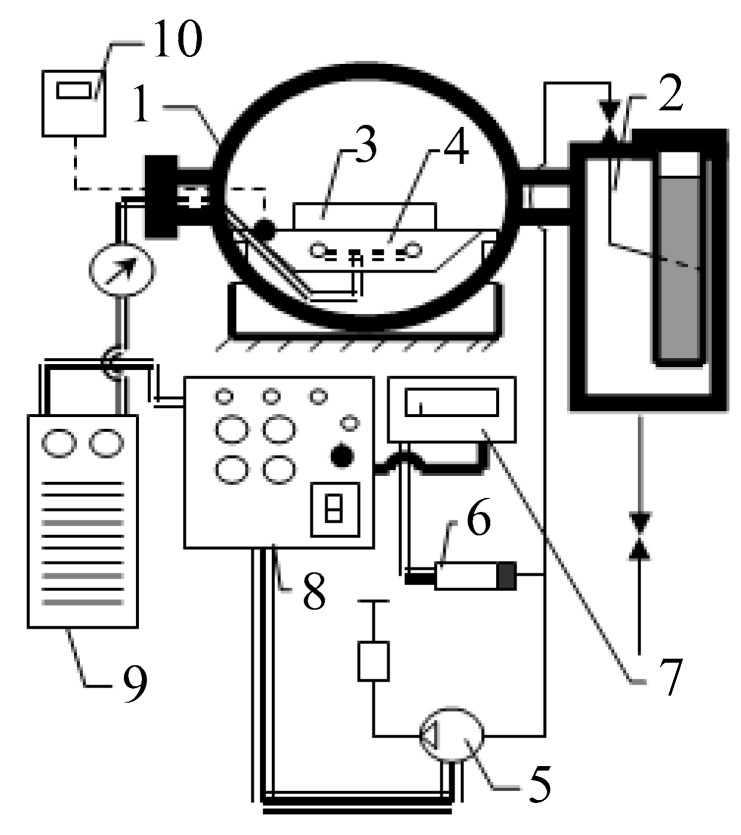
Vacuum sublimation apparatus: 1—sublimation chamber; 2—desublimator; 3—tray with a product; 4—heating stove; 5—vacuum pump; 6—gauge thermocouple; 7—vacuum gauge; 8—control panel; 9—universal voltage regulator, 10—electronic thermometer.

**Figure 4 nanomaterials-10-00624-f004:**
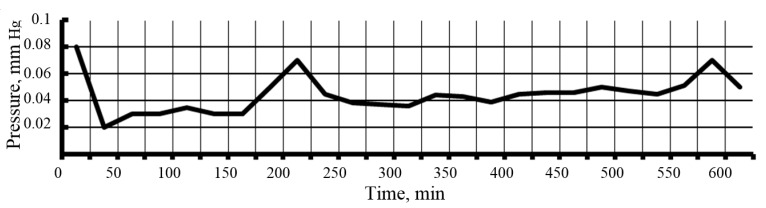
Dependence of pressure on time during sublimation upon receipt of a sample of nanopowder.

**Figure 5 nanomaterials-10-00624-f005:**
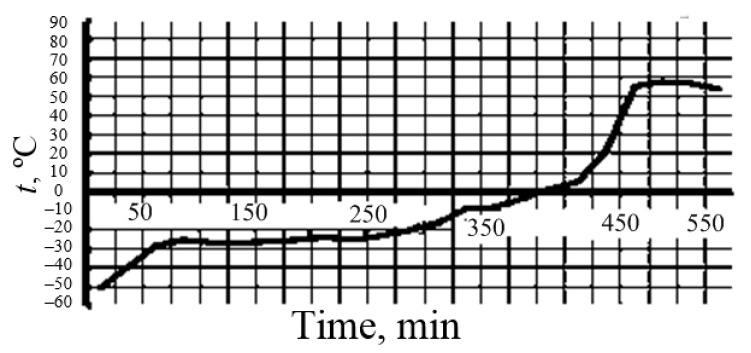
Dependence of temperature *t* in a vacuum chamber on the time of sublimation of cryogranules upon receipt of a nanopowder sample.

**Figure 6 nanomaterials-10-00624-f006:**
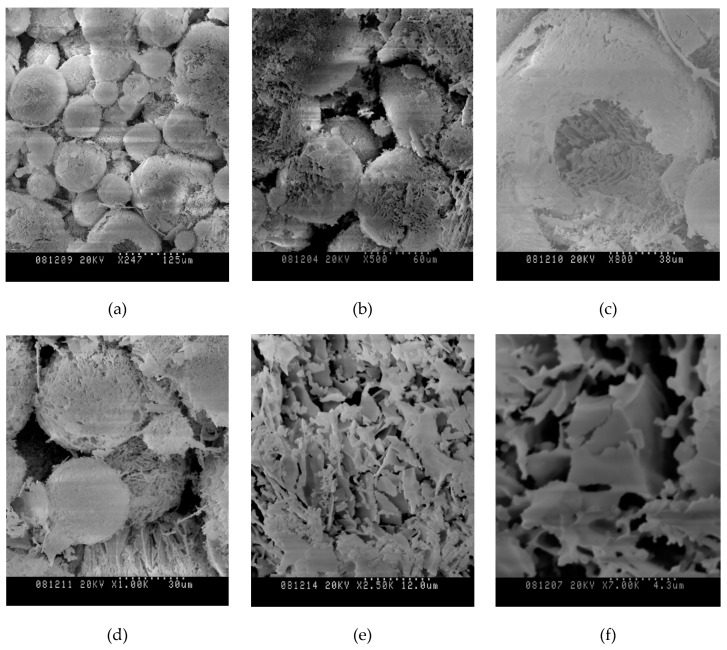
Images of structures from silica powder particles formed after sublimation of the solvent from cryogranules. The magnification factors on a scanning electron microscope: (**a**) 247 times; (**b**) 500 times; (**c**) 600 times; (**d**) 1000 times; (**e**) 2500 times; (**f**) 7000 times.

**Figure 7 nanomaterials-10-00624-f007:**
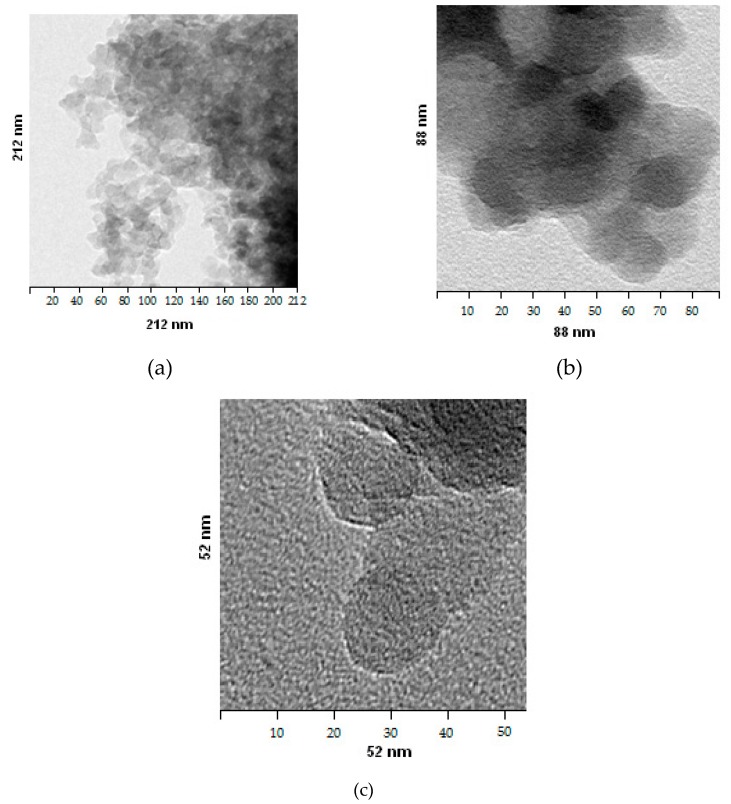
TEM images of SiO_2_ nanopowder particles obtained by tunneling electron microscopy. Dimensions of the image area of SiO_2_ particles: (**a**) 212 × 212 nm^2^; (**b**) 88 × 88 nm^2^; (**c**) 52 × 52 nm^2^.

**Figure 8 nanomaterials-10-00624-f008:**
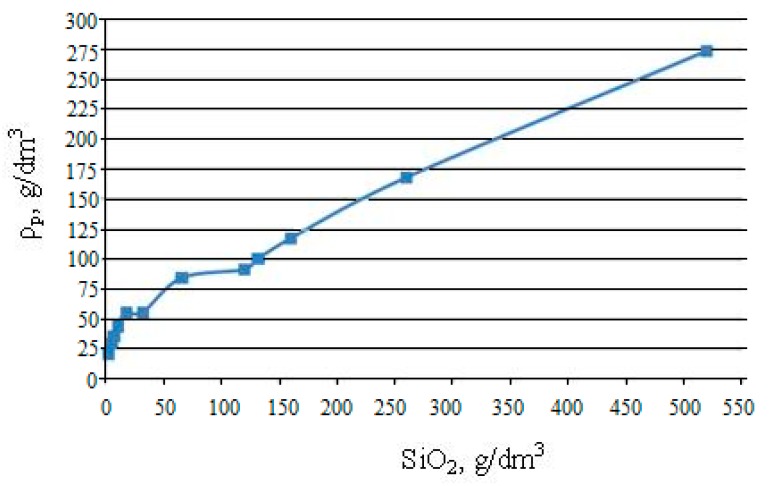
Dependence of the pour density of the nanopowder on the content of SiO_2_ in the sol.

**Figure 9 nanomaterials-10-00624-f009:**
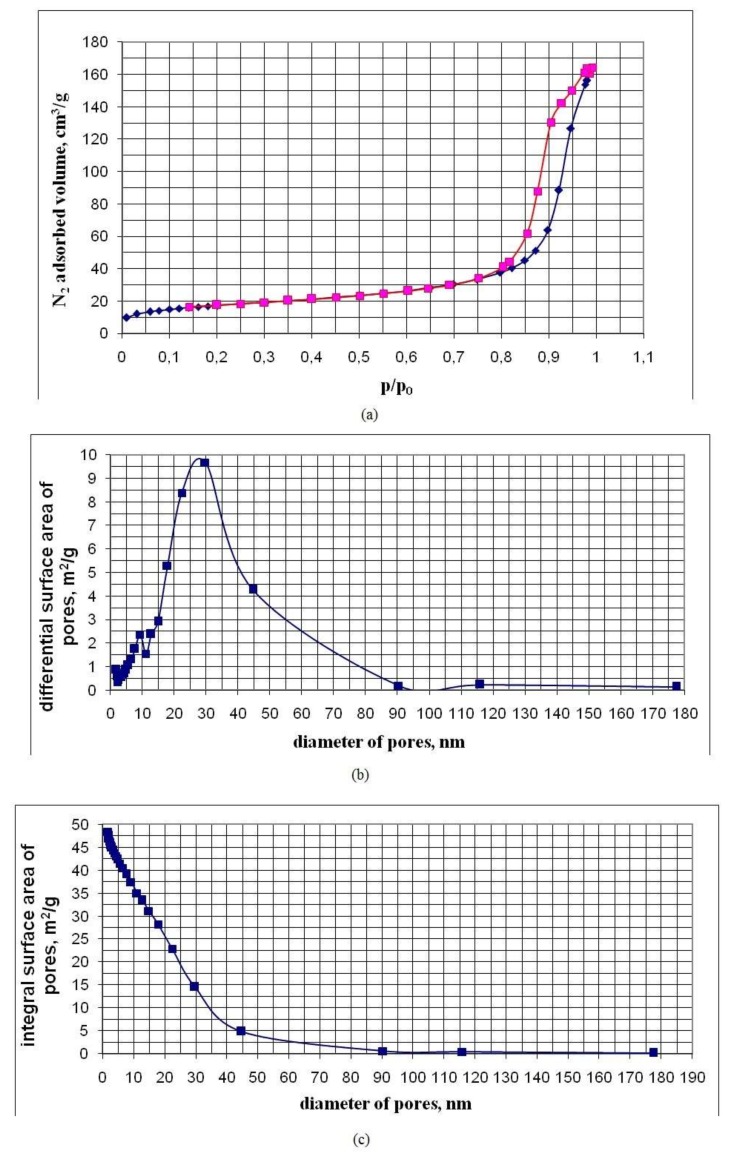
Pore characteristics of the UF-3-8 sample obtained by the low-temperature nitrogen adsorption method: (**a**) adsorption-desorption curves (p/p_0_—relative nitrogen pressure, p_0_—nitrogen saturation pressure at a temperature of 77 K); (**b**) differential distribution of area over pore diameter; (**c**) integral distribution of the area along the pore diameter; (**d**) xifferential distribution of volume over pore diameter; (**e**) integral distribution of volume by pore diameter.

**Figure 10 nanomaterials-10-00624-f010:**
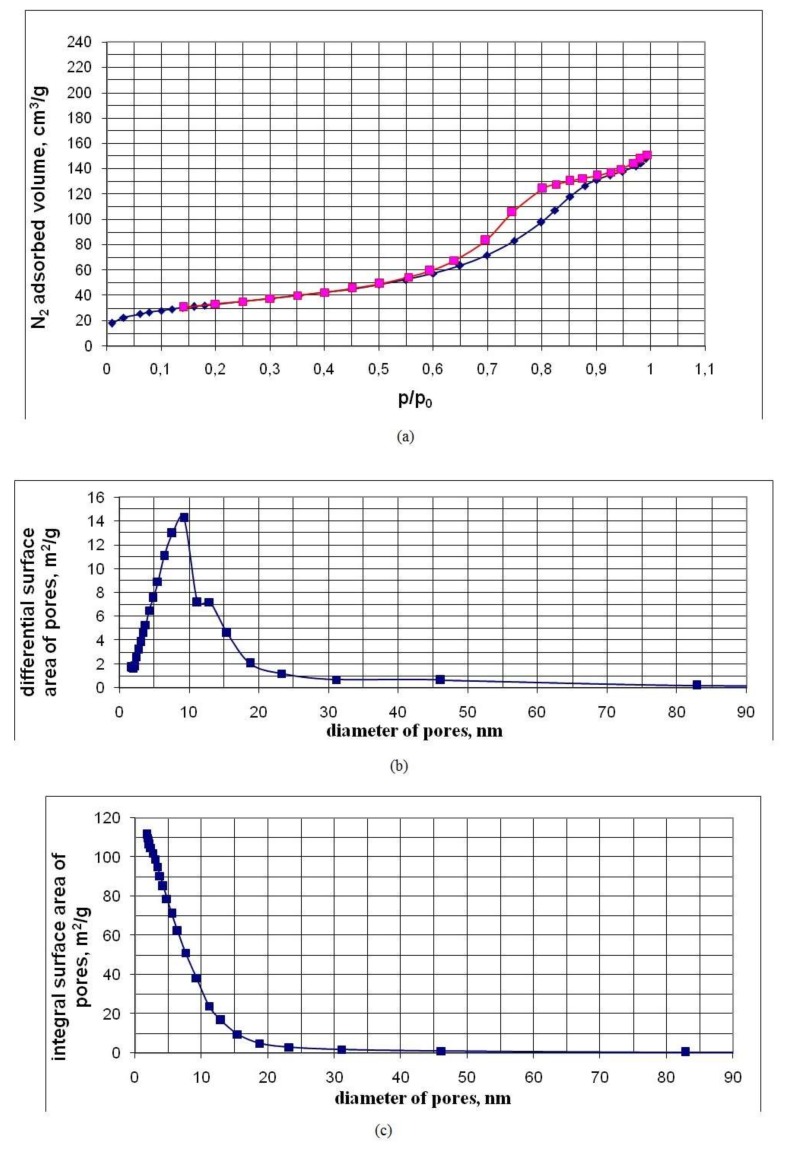
Pore characteristics of the UF-6-26 sample obtained by the low-temperature nitrogen adsorption method: (**a**) adsorption-desorption curves (p/p_0_—relative nitrogen pressure, p_0_—nitrogen saturation pressure at a temperature of 77 K); (**b**) differential distribution of area over pore diameter; (**c**) integral distribution of the area along the pore diameter; (**d**) differential distribution of volume over pore diameter; (**e**) integral distribution of volume by pore diameter.

**Figure 11 nanomaterials-10-00624-f011:**
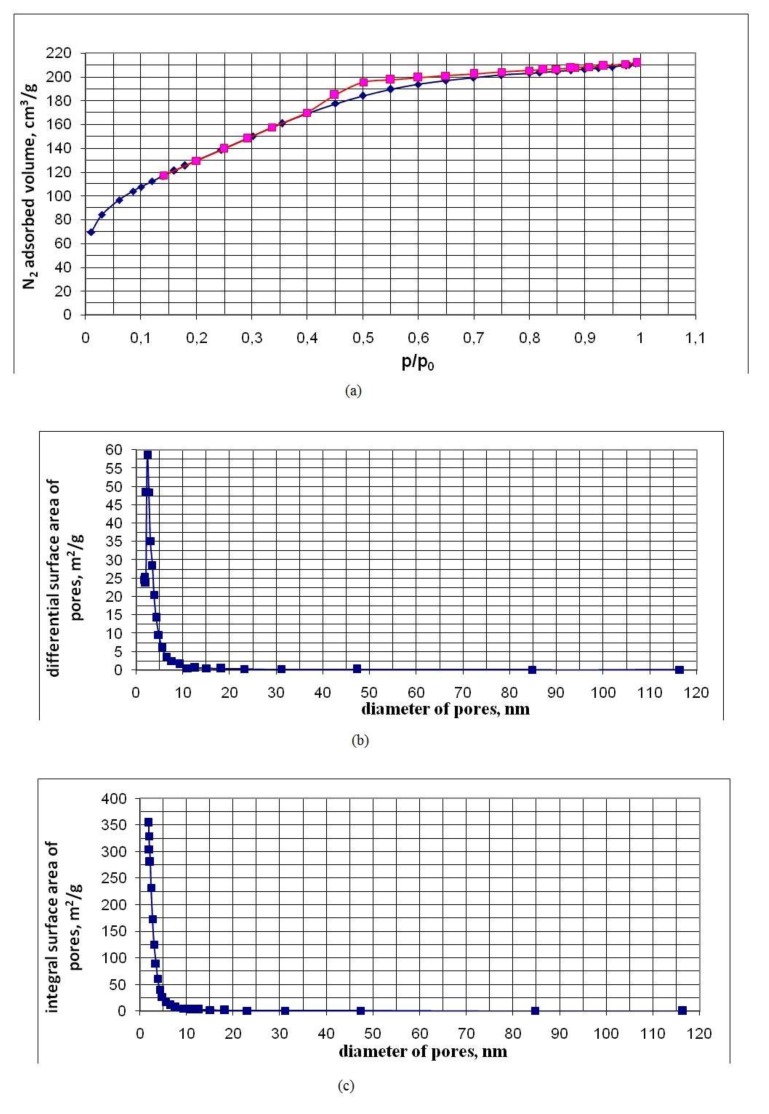
Pore characteristics of the UF-12-6 sample obtained by the low-temperature nitrogen adsorption method: (**a**) adsorption-desorption curves (p/p_0_—relative nitrogen pressure, p_0_—nitrogen saturation pressure at a temperature of 77 K); (**b**) differential distribution of area over pore diameter; (**c**) integral distribution of the area along the pore diameter; (**d**) differential distribution of volume over pore diameter; (**e**) integral distribution of volume by pore diameter.

**Figure 12 nanomaterials-10-00624-f012:**
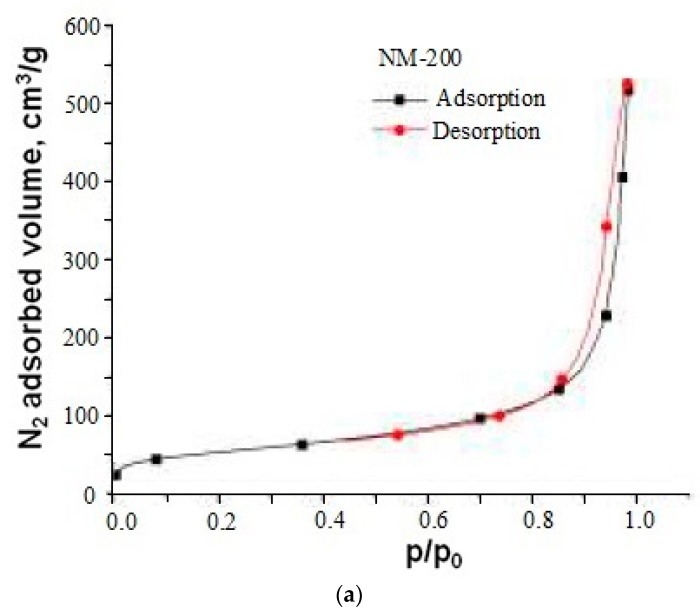
The curves of adsorption-desorption of precipitated NM-200, NM-201 and pyrogenic NM-202 samples of SiO_2_ nanopowders: (**a**) sample NM-200, (**b**) sample NM-201, (**c**) sample NM-202 [1].

**Figure 13 nanomaterials-10-00624-f013:**
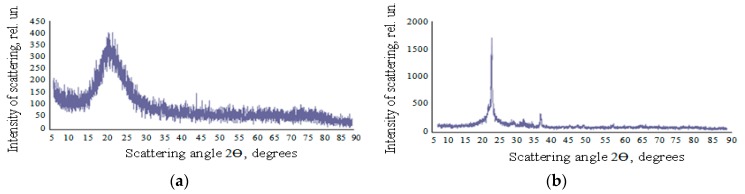
The XRD data of the nanopowder: (**a**) before calcination; (**b**) after calcination; Ө is the angle between the plane of the sample and the direction of radiation incidence. ARL X’TRA device (CuKa radiation, wavelength: 1.54 Å).

**Figure 14 nanomaterials-10-00624-f014:**
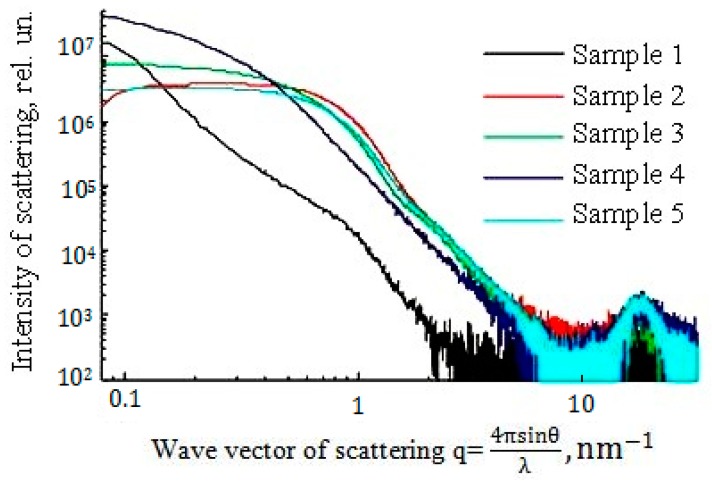
The results of measurements by the method of small-angle X-ray scattering.

**Figure 15 nanomaterials-10-00624-f015:**
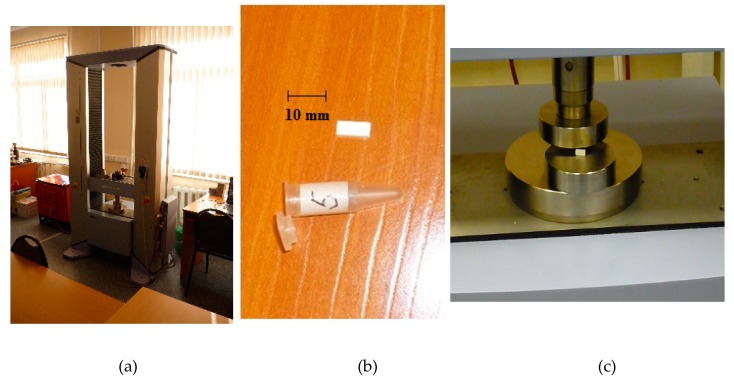
Strength tests of samples of compacted SiO_2_ nanopowder: (**a**) Shimadzu tester; (**b**) sample of compacted nanopowder; (**c**) sample before test.

**Figure 16 nanomaterials-10-00624-f016:**
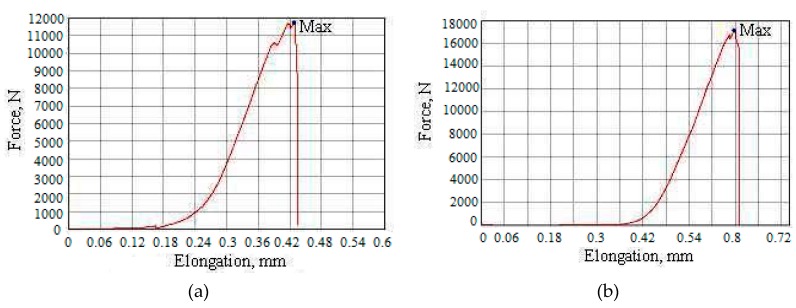
Curve force and elongation: (**a**) Sample 1; (**b**) Sample 2; (**c**) Sample 3; (**d**) Sample 4; (**e**) Sample 5.

**Table 1 nanomaterials-10-00624-t001:** The concentration of the main components of the initial hydrothermal solution.

Component	Na^+^	K^+^	Li^+^	Ca^2+^	Mg^2+^	Fe^2+, 3+^	Al^3+^	Cl^–^	SO_4_^2^	HCO_3_^–^	CO_3_^2–^	H_3_BO_3_	SiO_2 total_
Concentration, mg/dm^3^	282	48.1	1.5	2.8	4.7	< 0.1	< 0.1	251.8	220.9	45.2	61.8	91.8	780

Ionic strength of the solution *I_s_* = 14.218 mmol/kg, electrical conductivity *σ_el_* = 1.1–1.3 mS/cm, pH = 9.2.

**Table 2 nanomaterials-10-00624-t002:** Pour density of nanopowders, *ρ_p_*, depending on the SiO_2_ content in the sol.

**[SiO_2_],g/dm^3^**	2.4	5.2	6.93	10.4	17.56	32	65.85	120	131.7	160	260	520
***ρ_p_*, g/dm^3^**	20.5	29	35	43.8	55	55	84	91	100	117	168	274
**ρ_p_/[SiO_2_]**	8.5	5.6	5.05	4.21	3.13	1.71	1.27	0.76	0.76	0.73	0.65	0.53

**Table 3 nanomaterials-10-00624-t003:** Pore characteristics of powders established by low-temperature nitrogen adsorption.

Sample ID	[SiO_2_], g/dm^3^	ρ_p_, g/dm^3^	S_BET_, m^2^/g	Pore area by adsorption curve (BJH), S_BET,,_m^2^/g	Pore area by desorption curve (BJH), m^2^/g	Single point pore volume, v_p_,cm^3^/g	Pore volume by adsorption curve (BJH), cm^3^/g	Pore volume by desorption curve (BJH), cm^3^/g	d_BET_,nm	Average pore diameter, d_p_, nm	Average pore diameter by adsorption curve, nm	Average pore diameter by desorption curve, nm	Area of micropores, m^2^/g	Volume of micropores, cm^3^/g
UF-1-9	128.0	86	45.4	35.6	37.7	0.10	0.23	0.237	60	9.4	26.4	25.1	2.18	d.n.
UF-2-32	233.8	229	56.8	47.0	51.8	0.15	0.19	0.19	48.0	10.9	16.6	15.2	5.6	0.001
UF-3-8	24.4	52	62.0	48.3	58.0	0.19	0.24	0.25	44.0	12.6	20.5	17.3	11.4	0.004
UF-4-34	586.9	344	74.0	63.9	69.7	0.18	0.19	0.20	36.8	10.0	12.4	11.6	5.0	0.001
UF-5-25	108.9	52	97.7	78.8	90.4	0.22	0.26	0.27	27.9	9.4	13.5	11.9	13.9	0.005
UF-6-26	114.5	90	120.4	111.4	121.2	0.21	0.22	0.23	22.6	7.0	8.2	7.6	8.9	0.002
UF-7-17	28.0	35	166.5	151.4	162.1	0.25	0.28	0.28	16.4	6.2	7.5	7.1	8.2	0.001
UF-8-21	14.0	15.7	200.8	158.1	166.6	0.20	0.22	0.23	13.6	4.0	5.8	5.5	10.8	0.001
UF-9-43	170.9	231.7	209.9	199.6	239.1	0.21	0.20	0.22	13.0	4.0	4.0	4.0	0.1	d.n.
UF-10-3	82.5	90.0	316.0	272.1	289.9	0.243	0.216	0.221	8.6	3.0	3.2	3.0	d.n.	d.n.
UF-11-20	33.2	58	360.4	256.9	280.8	0.301	0.280	0.290	7.56	3.3	4.2	4.1	33.8	0.010
UF-12-16	66.0	86	476.3	354.3	367.1	0.32	0.26	0.27	5.72	2.70	3.0	2.94	0.1	d.n.

**Table 4 nanomaterials-10-00624-t004:** Pore characteristics of pyrogenic and precipitated SiO_2_ nanopowders [1] established by the BET(Brunauer–Emmett–Teller)-method.

Sample ID	ρ_p_, g/dm^3^	S_BET_, m^2^/g	Pore Volume, cm^3^/g	Area of Micropores, m^2^/g	Volume of Micropores, cm^3^/g
NM-200	120.0	189.1	0.79	30.0	0.01181
NM-201	280.0	140.4	0.581	23.1	0.00916
NM-202	130.0	204.1	0.513	8.26	0.00084
NM-203	30.0	203.9	0.499	5.3	0.0
NM-204	160.0	136.6	0.50	17.48	0.00666

**Table 5 nanomaterials-10-00624-t005:** The concentration of the chemical components of silica nanopowder (X-ray fluorescence spectrometer “S4 PIONEER”).

Oxides	Concentration, wt.%
SiO_2_	99.7
TiO_2_	0.00
Al_2_O_3_	0.173
FeO	0.00
Cr_2_O_3_	0.00
MgO	0.00
CaO	0.034
Na_2_O	0.034
K_2_O	0.069
MnO	0.00
NiO	0.00
ZnO	0.00
Total	100.0

**Table 6 nanomaterials-10-00624-t006:** Dependence of the mass of the nanopowder sample (wt.%) on temperature.

22.6 °C	100 °C	200 °C	300 °C	400 °C	500 °C	600 °C	700 °C	800 °C	900 °C	1000 °C	1100 °C
100%	94.65%	92.81%	92.10%	91.30%	90.58%	90.09%	89.76%	89.49%	89.27%	89.09%	88.61%

**Table 7 nanomaterials-10-00624-t007:** Distribution of OH-groups between surface and volume for hydrothermal silica sample.

***T*, °C**	200	300	400	500	600	700	800	900
**δ_OH_, OH/nm^2^**	8.29	6.71	4.92	3.33	2.23	1.49	0.89	0.40
**α_OH_, OH/nm^2^**	4.90	3.56	2.33	1.84	1.52	1.30	0.70	0.40
**γ_OH_, OH/nm^2^**	3.39	3.15	2.59	1.49	0.71	0.19	0.19	0.0

**Table 8 nanomaterials-10-00624-t008:** Characteristics of compacted SiO_2_ nanopowder samples during compressive strength tests.

Sample ID	Speed, mm/min	Shape	Dimensions (Thickness × Width × Height), mm	Maximum Force, N	Maximum Strain, N/mm^2^	Amplitude of the Stroke, mm	Maximum Elongation, %	Maximum Elongation, mm	Maximum Time, s
1	1	plane	4.9 × 11.9 × 3.3	11735.4	201.259	0.42831	12.9792	0.42831	25.7
2	1	plane	4.9 × 11.9 × 3.6	17145.4	294.040	0.65710	18.2529	0.65710	39.46
3	1	plane	5.5 × 13.5 × 3.1	10032.2	135.114	1.80619	58.2641	1.80619	108.370
4	1	plane	5.1 × 12.0 × 3.5	18897.6	308.784	1.81967	51.9905	1.81967	109.170
5	1	plane	5.0 × 11.9 × 2.9	20057.6	337.102	1.68196	57.9986	1.68196	100.950

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
