# Peer review of "Hydrothermal SiO2 Nanopowders: Obtaining Them and Their Characteristics"

_nanomaterials, 2020, doi:10.3390/nano10040624_

Round 1

Reviewer 1 Report

The technological process of the amorphous SiO2 nanopowder synthesis based on hydrothermal solutions is described.

Although I find this work interesting and potentially useful for journal readers some parts of this manuscript suffer from the editing. First of all some data could be collected into one plot, or moved to supporting information, i.e. Fig. 5, Fig. 10-13. There are too many plots in a raw.

Also some parts should be rewritten to avoid the report style rather than manuscript, i.e. conclusions.

Authors use the phrase "structure" referring rather to the silica morphology.

Overall the manuscript should be accepted only if the style of manuscript will be improved.

Author Response

 Figures 5 a, b, c, d were transformed in Figures 5 a, b.

Figures 10-13 were transformed in Figures 9, 10, 11. It is impossible to collect all the figures into one plot because the curves of different samples will not be well separated from each other.

Some parts of manuscript were changed: the aim of the work in Introduction, Conlcusion 1, results of comparison characteristics of hydrothermal nanosilica with another types of precipitated and pirogenic nanosilica were added (Figure 12, Table 4, the data of SAXS method).

The objectives of this article were:

- technological route for production of SiO2 nanopowders based on a hydrothermal solution with specific surface area up to 500 m2/g using the methods of ultrafiltration membrane separation and cryochemical vacuum sublimation;

- regulation parameters of the nanopowder’s structure: the diameters of SiO2 nanoparticles, specific surface area of nanopowders, diameters and specific pore volume, pour density, volume fraction of spherical particles in aggregates and agglomerates, sizes of agglomerates and number of particles in agglomerates.;

- assessment of possible applications of the obtained nanopowders.

Conclusions

  1. A technological route proposed that allows one to obtain amorphous SiO2 nanopowders based on a hydrothermal solution. The scheme includes OSA polycondensation processes, ultrafiltration membrane separation, and cryochemical vacuum sublimation. The route allows you to regulate parameters of the powder’s structure: the pour density, particle’s diameters, specific surface area, diameters, pore area and volume, volume fraction of spherical particles in aggregates and agglomerates, sizes of agglomerates and number of particles in agglomerates, fractal dimension. The parameters of the structure of hydrothermal nanosilica powders (ρp, dBET, SBET, Vp, Vs/Vaggr, 2∙Rg1, Dagglom, Nagglom, Df) differs from precipitated and pirogenic samples. The structure parameters determines physical and chemical activity and applications of nanopowders.

The term structure was used for such parameters as (ρp, dBET, SBET, Vp, Vs/Vaggr, 2∙Rg1, Dagglom, Nagglom, Df) . “Morfology” was used for the properties of the surface of SiO2 particles.    

Reviewer 2 Report

In this work, the authors reported on preparation and characterization of SiO2 nanoparticles synthesized by hydrothermal method. There are no novel ideas or significant conclusions were presented in this manuscript. In addition, overall physical mechanism of this manuscript is lack. Thus, I recommend this paper can’t be accepted for publication.

  1. Motivation of this work is not clear description.
  2. Some latest research results should be mentioned and cited in the introduction section instead of outdated or earlier papers.
  3. The corresponding scale bar should be included in Figure 7 and 16(a).
  4. Characters are too small in Figure 17.
  5. Physical properties of as-prepared nanoparticels do not compare with other works.

Author Response

Motivations of our work were to propose new technological route for nanosilica powder production with structure parameters differs from other types of nanosilica and to propose applications of this new type. The data were presented on physical and chemical mechanisms of OSA plycondensation, membrane separation and vacuum sublimation. Conclusions on structure parameters of new nanopowder were made. Applications for nanosilica were proposed based on experimental results.

The objectives of this article were:

- technological route for production of SiO2 nanopowders based on a hydrothermal solution with specific surface area up to 500 m2/g using the methods of ultrafiltration membrane separation and cryochemical vacuum sublimation;

- regulation parameters of the nanopowder’s structure: the diameters of SiO2 nanoparticles, specific surface area of nanopowders, diameters and specific pore volume, pour density, volume fraction of spherical particles in aggregates and agglomerates, sizes of agglomerates and number of particles in agglomerates.;

- assessment of possible applications of the obtained nanopowders.

A technological route proposed that allows one to obtain amorphous SiO2 nanopowders based on a hydrothermal solution. The scheme includes OSA polycondensation processes, ultrafiltration membrane separation, and cryochemical vacuum sublimation. The route allows you to regulate parameters of the powder’s structure: the pour density, particle’s diameters, specific surface area, diameters, pore area and volume, volume fraction of spherical particles in aggregates and agglomerates, sizes of agglomerates and number of particles in agglomerates, fractal dimension. The parameters of the structure of hydrothermal nanosilica powders (ρp, dBET, SBET, Vp, Vs/Vaggr, 2∙Rg1, Dagglom, Nagglom, Df) differs from precipitated and pirogenic samples. The structure parameters determines physical and chemical activity and applications of nanopowders.

We have added to the list of References more detailed information about results of research of different types of nanosilica by number of methods and have added the data of research of precipitated and pirogenic nanoslica by BET and SAXS methods for comparison with hydrothermal nanosilica (Figure 12, Table 4, 3.4 X-Ray and SAXS data). Sample NM-202 of pyrogenic SiO2 powder obtained by the flame hydrolysis of SiCl4 was characterized by SAXS, TEM, DLS and BET methods in [1]: Df =2.5, 2∙Rg1 = 16 nm, 2∙Rg2 = 100 nm, Nagglom = 200. Two samples NM-200 and NM-201 were produced by precipitation from precursor solution of Na2SiO3 were characterized by parameters:  NM-200 -  Df =2.45, 2∙Rg1 = 18 nm, 2∙Rg2 = 440 nm, Nagglom = 3500; M-201 - Df =2.45, 2∙Rg1 = 20 nm, 2∙Rg2 = 80 nm, Nagglom = 457. Fractal dimension of the hydrothermal nanopowders samples was lower than of the precipitated and pirogenic samples [1, 52, 54, 57, 58]. Physicochemical and biophysicochemical interactions of nanoparticles with cells are in strong dependence from fractal dimension Df and parameters of the structure of agglomerates 2∙Rg1, Dagglom, Nagglom, as from particles shape, surface electric charge and morphology [59-64].

  1. Beaucage, G., Small-Angle Scattering from Polymeric Mass Fractals of Arbitrary Mass- Fractal Dimension // J. Appl. Crystal. 1996. 29 (2). p. 134-146.
  2. Kammler, H. K.; Beaucage, G.; Mueller, R.; Pratsinis, S. E., Structure of Flame-Made Silica Nanoparticles by Ultra-Small-Angle X-ray Scattering // Langmuir. 2004. 20 (5). p. 1915-1921.
  3. Kammler, H. K.; Beaucage, G.; Kohls, D. J.; Agashe, N.; Ilavsky, J., Monitoring simultaneously the growth of nanoparticles and aggregates by in situ ultra-smallangle X-ray scattering // J. Appl. Physics. 2005. 97 (5). 054309-11.
  4. Hyeon-Lee, J.; Beaucage, G.; Pratsinis, S. E.; Vemury, S., Fractal Analysis of Flame- Synthesized Nanostructured Silica and Titania Powders Using Small-Angle X-ray Scattering // Langmuir. 1998. 14 (20). p. 5751-5756.
  5. Bushell, G. C.; Yan, Y. D.; Woodfield, D.; Raper, J.; Amal, R., On techniques for the measurement of the mass fractal dimension of aggregates. Advances in Colloid and Interface Science // 2002. 95 (1). p. 1-50.
  6. Brasil A.M., Farias T.L. and Carvalho M.G. A recipe for image characterization of fractal-like aggregates // Journal of Aerosol Science. 1999. 30(10). p. 1379-1389.
  7. De Temmerman, P.-J., Van Doren E., Verleysen E., Van der Stede Y., Francisco M.A.D. and Mast J. Quantitative characterization  of  agglomerates  and  aggregates  of  pyrogenic  and pprecipitated amorphous silica nanomaterials by transmission electron microscopy // Journal of       2012. 10. p. 24.
  8. Boldridge D. Morphological Characterization of Fumed Silica Aggregates. Aerosol Science and Technology // 2010. 44(3). p. 182-186.
  9. Nel A.E., Madler L., Velegol D., Xia T., Hoek E.M., Somasundaran P., Klaessig F., Castranova V., Thompson M. Understanding biophysicochemical interactions at the nano-bio interface // Nature Materials. 2009. 8(7). p. 543-557.
  10. Chu ,  Huang  Y.,  Tao  Q., Li  Q.  Cellular  uptake,  evolution,  and  excretion  of  silica nanoparticles in human cells // Nanoscale.  2011.  3(8). p. 3291-3299.
  11. Jiang ,  Oberdörster  G.  and  Biswas  P.  Characterization  of  size,  surface  charge,  and agglomeration  state  of  nanoparticle  dispersions  for  toxicological  studies // Journal of Nanoparticle Research. 2009. 11(1). p. 77-89.
  12. Powers W.,  Brown  S.C.,  Krishna  V.B.,  Wasdo  S.C.,  Moudgil  B.M.  and  Roberts  S.M.  Research  Strategies  for  Safety  Evaluation  of  Nanomaterials.  Part  VI. Characterization  of Nanoscale Particles for Toxicological Evaluation // Toxicol Sci. 2006. 90(2). p. 296-303.
  13. Roebben, G., Rasmussen, K., Kestens, , Linsinger, T. P. J., Rauscher, H., Emons, H., Stamm, H. Reference  materials  and  representative  test  materials:  the  nanotechnology  case // Journal of Nanoparticle Research. 2013. vol. 15. p. 1455-1468.
  14. Xia T., Kovochich M., Brant J., Hotze M., Sempf J., Oberley T., Sioutas C., Yeh J.I., Wiesner M.R., Nel, E.  Comparison  of  the  abilities  of  ambient  and  manufactured  nanoparticles  to induce cellular toxicity according to an oxidative stress paradigm // Nano Letters. 2006.  6/8. p. 1794-1807.

Scale bar was included in Figures 7 and 15.

Characters were magnified in Figure 16.

Reviewer 3 Report

The paper submitted is in general interesting, but in my opinion the manuscript is too long, the references are well chosen, the analyses proposed are appropriately selected. What seems to be missing is the list of reagents used and description of analytical techniques (should be described with sufficient details).

The entire manuscript should be checked again, the frequent deficiencies of space characters between words (especially after compound formulas) are particularly annoying. Also mathematical expression and units need to be checked (sometimes there are written with space characters, sometimes without), Celsius degree symbol is also written in two ways.

 As I mentioned before, the manuscript is too long, some of the figures could be transferred to the Supplementary Materials, especially the figures 9-13.

Author Response

We have tried to propose the technological way nanosilica production with out using chemical reagents. Production is based on initial hydrothermal solution, ultrafiltration membrane separation. We did not use analytical techniques.

Manuscript was checked on space characters, mathematical expressions, symbols. Corrections were made in the text of manuscript by red colour. 

Figures 9-13 were transformed in more compacted way.

Round 2

Reviewer 2 Report

The authors addressed the reviewers’ comments. 

I recommend the revision can be accepted for publication.